# Class Incremental Learning from First Principles: A Review

**Neil Ashtekar** *nca5096@psu.edu*
*Artificial Intelligence Research Laboratory*
*Pennsylvania State University*

**Jingxi Zhu** *jqz5678@psu.edu*
*Artificial Intelligence Research Laboratory*
*Pennsylvania State University*

**Vasant G Honavar** *vhonavar@ist.psu.edu*
*Artificial Intelligence Research Laboratory*
*Pennsylvania State University*

**Reviewed on OpenReview:** *https: // openreview. net/ forum? id= sZdtTJInUg*

## Abstract

Continual learning systems attempt to efficiently learn over time without forgetting previously acquired knowledge. In recent years, there has been an explosion of work on continual learning, mainly focused on the *class-incremental learning* (CIL) setting. In this review, we take a step back and reconsider the CIL problem. We reexamine the problem definition and describe its unique challenges, contextualize existing solutions by analyzing non-continual approaches, and investigate the implications of various problem configurations. Our goal is to provide an alternative perspective to existing work on CIL and direct attention toward unexplored aspects of the problem.

## 1 Introduction

In the past decade, machine learning has made tremendous progress on wide range of applications including computer vision, natural language processing, recommender systems, robotics, and more. The dominant approach – deep learning – generally performs well given large datasets in the i.i.d. setting, in which many passes may be made over the same data during training. However, existing methods fail when new data arrives, as users must either (i) store the old data and retrain jointly on both the old and new data, thus incurring significant memory and computation costs, or (ii) choose between learning from the new data and retaining previously learned knowledge. The trade-off described in (ii) is commonly known as the *stability-plasticity dilemma* (Carpenter & Grossberg, 1988). Recent work on continual learning has focused on ways to balance the stability-plasticity dilemma, with many solutions proposed to alleviate *catastrophic forgetting*, referring to the loss of old knowledge when learning anew (McCloskey & Cohen, 1989).

In this review, we consider supervised classification problems in the continual setting. Specifically, we focus on the *class-incremental learning* (CIL) problem setting, in which new classes of data are introduced over time (Van de Ven & Tolias, 2018). We seek to thoroughly understand both the problem statement and existing solutions, with a focus on unexplored directions.

Our work differs from recent continual learning review articles in several key ways, as summarized in Table 1. First, we specifically address the CIL setting, unlike De Lange et al. (2021); Mundt et al. (2023); Wang et al. (2024), and Verwimp et al. (2024). Second, we provide a novel categorization of existing approaches based on the shortcomings of non-continual methods, thus motivating and contextualizing recent work. Third, and perhaps most importantly, we focus on understanding and refining the CIL problem statement. This differs from Belouadah et al. (2021); Mai et al. (2022); Masana et al. (2022) and Zhou et al. (2024), which

Table 1: Brief summary of review articles on continual learning (CL), class-incremental learning (CIL), and task-incremental learning (TIL).

| Review | Submission Year | Content |
|---|---|---|
| De Lange et al. (2021) | 2019 | Reviews and evaluates TIL approaches, focused on relatively early work |
| Belouadah et al. (2021) | 2020 | Summarizes work on CIL with empirical evaluations on image classification benchmarks |
| Mai et al. (2022) | 2020 | Focuses on empirical evaluations in the online CIL setting over various performance metrics |
| Masana et al. (2022) | 2020 | Reviews work on CIL in the context of image classification with evaluations across various task-splits and replay strategies |
| Mundt et al. (2023) | 2020 | Summarizes work on CL and establishes connections to open-set recognition and active learning |
| Zhou et al. (2024) | 2023 | Reviews deep learning approaches to CIL with memory-aligned evaluations on image classification benchmarks |
| Wang et al. (2024) | 2023 | Broad overview of work on CL covering computer vision, natural language processing, reinforcement learning, etc. |
| Tian et al. (2024) | 2023 | Summarizes and evaluates CIL methods specifically designed for the few-shot setting (FSCIL) |
| Verwimp et al. (2024) | 2023 | Outlines real-world CL applications in which "continual learning is not a choice" and describes general directions for future work |
| **Ours** | **2024** | **Defines the CIL problem statement under resource constraints, describes properties of successful solutions** |

summarize and evaluate existing solutions. Instead, we investigate the challenging aspects of the problem and analyze the properties of a successful CIL system. Our main contributions are outlined below:

- In Section 2, we argue that the CIL problem should be redefined using constraints on memory and compute. We show that a commonly used problem statement is imprecise and admits extremely inefficient solutions.

- In Section 3, we describe key challenges introduced by CIL. We show that addressing these challenges is both necessary and sufficient for solving the broader problem. In particular, we highlight the importance of *across-task discrimination* and provide a corresponding performance metric.

- In Sections 4 and 5, we consider "naive" approaches which were not designed for continual learning. We analyze their strengths, weaknesses, and required modifications in the CIL setting, thus motivating existing work. Within this framing, we provide an overview of existing work.

- In Section 6, we investigate the dimensions of the CIL problem. We characterize when the problem is easy versus hard, and describe how existing work addresses various problem configurations. Notably, we investigate *class-to-task assignment* and provide a surprising example illustrating its importance.

- In Section 7, we conclude with suggestions for future work. Specifically, we argue that future solutions can and should make assumptions regarding the data generating process, as certain aspects of the world will remain fixed as new tasks are introduced.

## 2 Problem Statement

In this section, we formally define our problem statement. We focus our discussion on the class-incremental learning (CIL) setting as proposed in Van de Ven & Tolias (2018). We borrow notation and definitions from Zhou et al. (2024) and Kumar et al. (2023). First, we state the objective of CIL, then we provide two forms of restrictions imposed on potential solutions. These restrictions are crucial for ensuring that the problem is well defined and that solutions are non-trivial (i.e., CIL could be trivially solved by storing and repeatedly retraining on all of the data, though this is highly inefficient). We argue that restrictions in the form of *resource constraints* are more appropriate than restrictions on the availability of previously learned data.

### 2.1 Class-Incremental Learning

In CIL, we wish to sequentially learn a set of tasks $1, \dots, T$. For each task $k \in \{1, \dots, T\}$, we are given a corresponding dataset $\mathcal{D}_k = \{(\boldsymbol{x}_i^k, y_i^k)_{i=1}^{n_k}\}$ sampled from an unknown, underlying distribution $\mathcal{D}_k \sim \mathcal{T}_k$. Each task is a supervised classification problem with feature vectors $\boldsymbol{x}_i^k \in \mathbb{X}$ and categorical labels $y_i^k \in \mathbb{Y}_k$. One or more classes are included in each task, and classes are disjoint across tasks ($\mathbb{Y}_k \cap \mathbb{Y}_{k'} = \emptyset$, $\forall k \neq k'$ and $\cup_{k=1}^{T} \mathbb{Y}_k = \mathbb{Y}$).

The overall goal is to learn a classifier $f : \mathbb{X} \to \mathbb{Y}$ with strong generalization performance across all classes. Task-identification information is not available to the model at inference. The classifier should perform well on all previously observed classes after learning each task: after learning task $k$, performance should be strong on all classes $\mathbb{Y}_{1:k}$. At task $k$, this goal can be expressed as learning $f_k^*$ from the model's hypothesis space $\mathcal{H}$ such that:

$$f_k^* = \operatorname*{argmin}_{f_k \in \mathcal{H}} \mathbb{E}_{(\boldsymbol{x}, y) \sim \mathcal{T}_{1:k}} [\mathbb{I}(f_k(\boldsymbol{x}) \neq y)] \tag{1}$$

### 2.2 Restrictions while Learning

#### 2.2.1 Old Restriction: Unavailability of Previous Data

Much of the existing work on CIL assumes that the data from previous tasks is not available when learning the current task. Formally, when learning task $k$, the model can only access $\mathcal{D}_k$, while $\mathcal{D}_{1:k-1}$ are inaccessible. Many approaches to CIL (Kim et al., 2022; Van De Ven et al., 2021; Zhu et al., 2021a; Zhang et al., 2020a; Tao et al., 2020) as well as a recent survey on deep CIL (Zhou et al., 2024) explicitly include this restriction in the problem statement.

Completely restricting access to previous task data may appear reasonable. This restriction disallows trivial approaches which retrain on all of the data at every task. Also, security and/or privacy concerns are often cited as a reason to restrict access to old data. However, we argue that this restriction is both *unnecessary* and *inadequate* for building useful CIL systems. To understand why, consider the following critiques.

**Critique #1: Inefficient solutions are allowed.** First, if the unavailability of previous data is the only restriction (i.e., there are no other restrictions on memory and/or compute), then extremely inefficient solutions are allowed. In a recent analysis on the ImageNet-1K dataset, Harun et al. (2023a) finds that several highly-cited CIL algorithms actually use more compute[1] than trivially retraining on all of the data at every task! In one respect, such algorithms defeat the purpose of using continual learning.

**Critique #2: Security and privacy violations may persist.** Second, note that deep neural networks have the capacity to essentially "memorize" training data (Zhang et al., 2021) and that decoding schemes can be used to reconstruct data from a trained network (Haim et al., 2022). As discussed in Verwimp et al. (2024), these observations lead us to question whether restricting access to old data truly addresses security or privacy concerns, as sensitive data could simply be recovered from trained models. This concern is not specific to deep neural networks – even simple generative or prototype-based classifiers may memorize data or be susceptible to data leakage.

---

[1] Measured in total number of backpropagation updates

**Critique #3: Imprecise problem definition.** Third, if previous task data cannot be stored but previously learned model(s) can be stored, then it becomes necessary to formally answer the question "What is a model?" or at least answer "What is the difference between a model and a dataset?" The existence of models such as K-nearest neighbors (which store the entire dataset) and prototype-based classifiers (which store representatives which may be very similar to the data) make these questions difficult to answer. Note that some early work has defined learning algorithms (models) with respect to data compression in the PAC learning framework (Blumer et al., 1987; Takimoto & Maruoka, 1993), though these definitions have not been referenced in the modern continual learning literature. For this reason, we argue that framing the CIL problem based on the unavailability of previous data results in an imprecise problem definition.

*A note regarding relaxations of this restriction.* We acknowledge that many approaches to CIL consider a relaxation of this restriction in which a relatively small subset of previous task data is available when learning the current task (see the discussion on replay-based approaches in Section 5). In this framing, solutions are typically allowed to store either a fixed number of samples per-class (Rolnick et al., 2019) or a fixed number of total samples (Wang et al., 2023). While we feel that this framing is a step in the right direction, we argue that it is still lacking. This is because (i) it does not include restrictions on compute, (ii) the number of stored samples is constrained but the size of the predictive model is not, and (iii) unnecessary assumptions are made regarding data storage. Drawback (i) allows computationally-inefficient solutions, which may be undesirable for CIL applications. Drawback (ii) implies that the total memory cost – stored samples *and* model parameters – may not be appropriately measured or constrained. Finally, drawback (iii) may introduce artificial constraints limiting solution performance. For example, it may be suboptimal to store the same number of samples for each class. Alternatively, storing task-specific model components can sometimes be a better use of a given memory budget as compared to storing data samples, as discussed in Zhou et al. (2023).

### 2.2.2 New Restriction: Resource Constraints

An alternative framing of the CIL problem restricts solutions using resource constraints instead of restricting access to previous task data. Recent work frames the continual learning problem as computationally constrained reinforcement learning (Kumar et al., 2023). In this framing, a continual learning agent attempts to maximize its average reward[2] under restrictions on memory and compute. For example, memory constraints may limit the total size of a model, while computational constraints may limit the number of floating point operations performed when learning each task.

Resource constraints prevent trivial retraining using all of the data at every task, either because storing the entire dataset would require too much memory and/or because full retraining would be too computationally expensive. Because trivial retraining is infeasible, continual learning methods are necessary. We argue that framing the CIL problem in terms of resource constraints rather than data availability results in a more appropriate problem statement and will lead to the development of truly useful, real-world CIL systems.

Resource constraints can take many forms. For example, edge devices performing CIL may have strict latency or power constraints, permitting only a small number of learning updates (Yoshikiyo et al., 2022; Wang et al., 2022b). On the other hand, large GPU servers performing CIL may be relatively unconstrained with respect to memory, and are instead constrained by the monetary cost of compute (e.g., renting large instances on cloud computing platforms may be prohibitively expensive, see Prabhu et al. (2023b) for details). In this case, it may be desirable to develop CIL algorithms which jointly optimize for classification performance as well as computational efficiency. Here, resource constraints are not fixed, but rather become part of the optimization objective.

In summary, we emphasize the importance of including application-specific resource constraints as part of the CIL problem definition. Doing so avoids the issues caused by restricting the availability of previous task data: resource constraints formally define the space of potential solutions and disallow loopholes, there is no need to formally define "model" versus "data", and the problem is not made artificially difficult by misconstrued privacy or security concerns. In the following sections, we will consider the resource-constrained version of the CIL problem statement.

---

[2]In CIL, average reward is equivalent to average negative log-loss (i.e., cross-entropy loss) over all classes.

### 2.3 Related Problem Statements

We briefly describe other continual classification problem statements below, based on the categorization given in Van de Ven & Tolias (2018). While the focus of this work is CIL, note that the arguments for the resource-constrained setting in Section 2.2 are applicable to CIL as well as TIL, DIL, and OCL, and extend to continual learning problem settings even beyond classification.

**Few-Shot Class-Incremental Learning (FSCIL).** FSCIL involves continually learning new classes of data given only a small number of samples per class. This setting is typically addressed using some form of *pretraining*: learning on a large, general-purpose dataset with many samples prior to continual learning. FSCIL inherits the challenges of CIL described in Section 3.1, with the added difficulty of *unreliable empirical risk minimization*, i.e., avoiding overfitting when learning from a small sample size (Wang et al., 2020).

**Task-Incremental Learning (TIL).** The TIL problem setting is very similar to that of CIL: new classes are introduced incrementally, and the classifier must efficiently adapt while maintaining previously learned knowledge. However, in TIL, *task-identifiers are provided at inference*. This removes the need for learning across-task discrimination, discussed in Section 3.1. TIL can be viewed as an "easier" version of CIL.

**Domain-Incremental Learning (DIL).** In DIL, the set of classes is fixed, but the distribution of features for each class changes over time. Similarly to CIL and TIL, the classifier must continually learn while mitigating forgetting. Unlike CIL and TIL, this challenge must be addressed at the level of feature distributions rather than at the class-level. Task-identifiers are unavailable at inference. Therefore, the key challenges described in Section 3.1 for CIL each have counterparts for DIL.

**Online Continual Learning (OCL).** OCL involves learning from small batches of data introduced incrementally (Mai et al., 2022). This differs from (non-online) CIL, in which *all* of the data from a given class is available when learning its corresponding task. In other words, samples from a single class may be split across multiple timesteps. In the extreme case, samples may be introduced one at a time. This introduces the additional challenge of aggregating knowledge across timesteps for samples within a given class. Note that OCL is a broad categorization which is not specific to classification. In addition, these problem settings can be combined. For example, CIL + DIL + OCL is sometimes referred to as *Task-Free Continual Learning* (TFCL) (Aljundi et al., 2019a).

### 2.4 Real-world CIL Example: Social Media Post Classification

To motivate our problem statement, we outline a potential real-world system which fits the resource-constrained CIL setting. We discuss the problem of *real-time, social media post classification*, with new classes introduced incrementally. Classes could correspond to post content (text, images, or videos) with applications for fraud detection, content moderation, targeted advertising, and more. Note that this problem may not be trivially solvable using hashtags or interest tagging, as posts may be implicitly referring to a given topic and fraudulent content may be intentionally obfuscated. The problem may be framed as "traditional" CIL or online CIL depending on how the data is collected and training is scheduled.

Popular social media platforms receive an extremely high volume of posts – according to Twitter's official blog, an average of 500 million tweets were shared every day in 2014 – and new classes may be introduced in a short time period corresponding to current events or newly developed types of fraud. Further, posts may need to be classified in near real-time in order to avoid the spread of misinformation or limit online scams. These concerns motivate the resource-constrained setting – the extremely high volume of posts necessitates computationally efficient learning, and the real-time nature of the application requires a low inference cost.

In such a setting, the entire sequence of training data is stored by default (i.e., posts from several years ago are available on a user's profile and therefore must be stored on the platform's server). This removes the "unavailability of previous data" restriction discussed in Section 2.2 and places on the focus instead on computational efficiency. Note that this shift in focus significantly changes the solution space. Solutions leveraging K-nearest neighbors or other forms of locally-weighted learning may be appropriate, and such techniques could be learned atop pretrained representations – see Prabhu et al. (2023b) for one such approach.

Such an approach may serve as a starting point, with modifications necessary to meet performance and efficiency goals. For example, the pretrained representation may need to be continually fine-tuned in order to improve classification accuracy, requiring some form of replay and/or knowledge distillation (see Section 5 for details). Further, even if all of the training data is stored, computational constraints may limit the amount of data used when learning continually, necessitating an intelligent sampling strategy. Finally, efficient inference techniques (e.g., forms of locality-sensitive hashing) may be required to meet near real-time classification constraints. Note that the example of social media post classification is not unique: applications ranging from cybersecurity to epidemiology involve similarly high volumes of data and may require fast inference. For further examples of real-world continual learning applications, see Verwimp et al. (2024).

## 3 Key Challenges

It is worth understanding what makes the CIL problem challenging as compared to learning in the non-continual setting. We outline three main challenges: balancing stability and plasticity, learning across-task discrimination, and transferring knowledge across tasks. None of these challenges are present in the non-continual setting, which can be thought of as a single task. We first describe these challenges then discuss why they are significant.

### 3.1 Challenges

**Balancing stability and plasticity.** The stability-plasticity dilemma refers to the ability to learn new tasks while maintaining previously learned knowledge (Carpenter & Grossberg, 1988). Loss of stability is typically referred to as *forgetting* – or *catastrophic forgetting* (McCloskey & Cohen, 1989), due to its empirically observed severity – and can be measured as the degradation of model performance on old tasks as new tasks are learned. Lack of plasticity can be defined using *intransigence* (Chaudhry et al., 2018a). Intransigence reflects a model's inability to acquire new knowledge, measured as the difference in performance across the continual and non-continual settings.

To provide formal definitions, we introduce notation[3] originally proposed in Lopez-Paz & Ranzato (2017). Let $\boldsymbol{R} \in \mathbb{R}^{T \times T}$ be the train-test performance matrix, with elements indicating some performance measure[4] for a given classifier. Element $\boldsymbol{R}_{i,j}$ denotes the performance on the test set of task $j$ immediately after learning the training set of task $i$. We assume that tasks are learned in the order in which they are presented: tasks $1, \ldots i-1$ are learned in order before task $i$ is learned. For all of the following metrics with subscripts $i$ and $j$, we assume that task $i$ is learned before task $j$. Formally, $i \leq j$ in Eqs. 2, 4, 5, and 6.

The amount of forgetting on task $i$ after learning tasks $i+1, \ldots, j$ can be quantified as:

$$\boldsymbol{F}_{i,j} = \max_{k \in \{i, \ldots, j-1\}} \boldsymbol{R}_{k,i} - \boldsymbol{R}_{j,i} \tag{2}$$

The max operation is included in order to account for backward knowledge transfer – see discussion following Eq. 6 for details. The intransigence on task $i$ after learning tasks $1, \ldots, i-1$ can be quantified as:

$$\boldsymbol{I}_i = \boldsymbol{R}_i^* - \boldsymbol{R}_{i,i} \tag{3}$$

Here, $\boldsymbol{R}_i^*$ denotes the performance of a reference model on task $i$. This reference model is learned in the non-continual setting, i.e., jointly trained on all data $\cup_{k=1}^{T} \mathcal{D}_k$. Performance in the non-continual setting is often used as an upper bound for performance in the continual setting, and the corresponding performance gap indicates an inability to continually learn new tasks. For fair evaluations, the reference model and continual learning model should have similar architectures.

---

[3]We provide simplified notation for pairs of tasks – more detailed notation for overall forgetting and knowledge transfer are given in Lopez-Paz & Ranzato (2017); Chaudhry et al. (2018b); Díaz-Rodríguez et al. (2018)

[4]Classification accuracy is commonly used as a performance measure. In general, measures should be chosen such that higher values correspond to better performance.

**Learning across-task discrimination.**[5] At inference, CIL methods must perform classification across all learned classes without the aid of task-identifiers. Namely, after learning task $k$, the model should be able to perform classification over all classes $\mathbb{Y}_{1:k}$. This requires the ability to discriminate between classes *within each task* and *across all tasks*. Note that the definitions of stability and plasticity defined above are insufficient for ensuring that across-task discrimination is learned. This is because these definitions only consider within-task performance. CIL solutions must include some mechanism to learn and maintain across-task discrimination.

To formally define across-task discrimination, consider the normalized confusion matrix $\boldsymbol{M}^j$ populated after learning tasks $1, \ldots, j$. Matrix $\boldsymbol{M}^j$ has dimensionality $|\mathbb{Y}_{1:j}| \times |\mathbb{Y}_{1:j}|$,[6] with element $\boldsymbol{M}_{m,n}$ indicating the proportion of instances of class $m$ which are predicted as class $n$ after evaluation on a test set. In other words, the rows of $\boldsymbol{M}^j$ correspond to the actual classes, while the columns of $\boldsymbol{M}^j$ correspond to the predicted classes. To simplify notation, let $\mathbb{C}_i$ be the set of indices corresponding to the classes in task $i$. For example, $\mathbb{C}_1 = \{1, \ldots, |\mathbb{Y}_1|\}$, $\mathbb{C}_2 = \{|\mathbb{Y}_1| + 1, \ldots, |\mathbb{Y}_1| + |\mathbb{Y}_2|\}$, and so on. Within a task, the assignment of indices to classes is arbitrary. Consider a model which has learned tasks $1, \ldots, j$, and consider a task $i$ such that $i \leq j$. The model's across-task discrimination performance with respect to task $i$ can be defined as:

$$\boldsymbol{ATD}_{i,j} = \frac{1}{|\mathbb{C}_i|} \sum_{m \in \mathbb{C}_i} \sum_{n \in \mathbb{C}_i} \boldsymbol{M}^j_{m,n} \tag{4}$$

This quantity simply measures how often the model predicts *a class from the correct task* at inference. Visually, it corresponds to the values within a square subset of the confusion matrix along the diagonal.

**Transferring knowledge across tasks.** It is reasonable to assume that there will be similarity across tasks in a given CIL application. Ideally, knowledge accumulated from old tasks should improve performance when learning new tasks, and learning new tasks should also improve performance on old tasks. In other words, both *forward transfer* and *backward transfer* are desirable. Consider two tasks, $i$ and $j$, with $i < j$. The forward transfer from earlier tasks $1, \ldots, i$ to later task $j$ can be measured as:

$$\boldsymbol{FWT}_{i,j} = \boldsymbol{R}_{i,j} \tag{5}$$

This definition of forward transfer is equivalent to the model's *zero-shot* performance on task $j$. The backward transfer from later tasks $i+1, \ldots, j$ to earlier task $i$ can be measured as:

$$\boldsymbol{BWT}_{i,j} = \max_{k \in \{i, \ldots, j-1\}} \boldsymbol{R}_{j,i} - \boldsymbol{R}_{k,i} \tag{6}$$

This definition of backward transfer is similar to the negation of forgetting in Eq. 2. Forgetting measures how much performance on an earlier task *decreases* when learning later tasks, while backward transfer measures how much performance on an earlier task *increases* when learning later tasks. These metrics are typically reported only when their values are nonnegative: when backward transfer is positive, it is reported rather than "negative forgetting", and vice versa. Note that Eqs. 2 and 6 both include the max operation – this is because both definitions are cumulative. For example, knowledge may be transferred backward and later forgotten, or knowledge may forgotten and later re-learned through backward transfer. Eqs. 2 and 6 account for such situations. Forgetting and backward transfer could alternatively be defined without the max operation (replacing the $\boldsymbol{R}_{k,i}$ term with $\boldsymbol{R}_{i,i}$) resulting in a non-cumulative definitions.

## 3.2 Why these challenges?

**Successful within-task discrimination is a consequence of stability and plasticity.** While within-task discrimination is a necessary component of successful CIL systems, we do not include it as a key

---

[5]Across-task discrimination is sometimes called cross-task discrimination, inter-task separability, or inter-task confusion (Masana et al., 2022; Huang et al., 2023; Nori & Kim, 2024)

[6]We use vertical bars to indicate the number of elements within a set (cardinality). Here, $|\mathbb{Y}_{1:j}|$ is the total number of classes learned in tasks $1, \ldots, j$

challenge. Instead, we argue that achieving and maintaining strong within-task performance is a consequence of sufficient stability and plasticity. Recall that $R_{i,j}$ denotes the performance on task $j$ after learning task $i$. Strong within-task performance can be formalized as follows: immediately after task $i$ is learned, $R_{i,j}$ should be sufficiently high for all previously learned tasks $j \leq i$. This should hold for all tasks $i \in \{1, \ldots, T\}$. Note that the terms "strong performance" and "sufficiently high" are subjective, and can be defined based on application-specific and/or task-specific criteria.

This formalization simply requires that performance be strong for the most recently learned task as well as for all of the other previously learned tasks. Strong performance on the most recently learned task is equivalent to low intransigence in Eq. 3. Maintaining performance on prior tasks is equivalent to low forgetting in Eq. 2. Therefore, sufficient stability and plasticity results in successful within-task discrimination.

**Learning across-task discrimination is necessary for CIL.** As proved in Kim et al. (2022), strong within-task discrimination and strong across-task-discrimination are together *necessary* and *sufficient* conditions for strong CIL performance. In other words, a model will have strong overall performance if and only if it is can accurately distinguish between classes from the same task and across classes from different tasks. Learning and maintaining across-task discrimination is therefore a key challenge in CIL.

Strong across-task performance can be formalized as follows. Immediately after learning task $j$, $ATD_{i,j}$ (Eq. 4) should be sufficiently high for all previously learned tasks $i \leq j$. This condition should hold for each task $j \in \{1, \ldots, T\}$. This formalization is very similar to the formalization of strong within-task performance, with $ATD_{i,j}$ used instead of $R_{i,j}$. However, there is an important difference between the within-task and across-task discrimination problems: *the across-task problem grows as new tasks are introduced, while each within-task problem has a fixed number of classes.* It may be necessary to recalibrate performance expectations as the number of tasks grows and across-task discrimination increases in difficulty.[7]

In addition, we argue that *across-task discrimination is a larger part of the overall CIL problem as compared to within-task discrimination.* To provide evidence for this claim, we first discuss a commonly used CIL benchmark – the split CIFAR-100 dataset – then discuss the general case. The CIFAR-100 dataset contains 100 classes, and is often divided[8] into 10 tasks, each containing 10 classes, to form the "split" version for CIL evaluations. Given a sample at inference after all tasks have been learned, the model should predict its corresponding class out of the 100 total classes. This requires distinguishing the correct class (i) versus 9 incorrect classes from the same task and (ii) versus 90 incorrect classes from other tasks. In this example, across-task discrimination dominates the CIL problem, as the model must predict across many more classes out-of-task versus within-task.

This observation extends beyond any specific dataset. Unless one task contains more than half of all of the learned classes, *it will always be true* that across-task discrimination is a larger part of the CIL problem as compared to within-task discrimination. While a couple of recent studies (Soutif-Cormerais et al., 2021; Guo et al., 2023) focus specifically on the across-task discrimination problem in CIL, we are not aware of any prior work which explicitly mentions this observation. This is noteworthy considering that many CIL approaches heavily emphasize the avoidance of within-task forgetting while placing relatively little emphasis on across-task discrimination – see Section 5 for details.

**Knowledge transfer can improve both performance and resource efficiency.** Exploiting knowledge transfer between similar tasks could improve a model's predictive performance and decrease both memory and computational costs. For example, sharing parameters across tasks could decrease overall model size compared to learning isolated submodels for each task. A model could exploit forward transfer (Eq. 5) by initializing new, task-specific parameters to the values learned on a prior, similar task, thus reducing the number of iterations required for convergence. Such approaches may be necessary to achieve a desired level of predictive performance under resource constraints.

---

[7]Across-task discrimination is generally harder given more tasks. For example, we would expect stronger performance when discriminating between two tasks as compared to ten tasks. See Michel et al. (2023) for difficulty-adjusted metrics for CIL.

[8]Class-to-task assignment is typically performed randomly or based on the sequential class labels, e.g., classes 0 - 9 are assigned to the first task, 10 - 19 assigned to the second class, etc.

# 4 Naive Approaches

In this section, we discuss the application of traditional (i.e., non-continual) machine learning classifiers in the CIL setting. This is "naive" in the sense that these models would be expected to fail in the CIL setting, as they are not designed for continual learning. We discuss three types of classifiers: (1) a single discriminative model for all tasks, (2) task-specific discriminative models, and (3) class-specific generative models. These approaches are illustrated in Figure 1.

Some of the observations in this section may seem obvious or redundant in light of recent work on continual learning. However, we feel that this section is important as it provides a principled way of developing solutions to the CIL problem. This section also serves to contextualize existing work and provide insights for potential future work. Table 2 and Table 3 summarize the main takeaways. Table 2 outlines how each naive approach addresses the "Key Challenges" described in Section 3, while Table 3 outlines the memory and compute requirements for each approach, relevant to the resource-constrained problem statement defined in Section 2.

## 4.1 Background: Generative and Discriminative Classifiers

**Discriminative Classifiers.** Probabilistic discriminative classifiers directly model $P(y|\boldsymbol{x})$. Discriminatively trained models such as logistic regression and deep learning classifiers typically accomplish this by optimizing for cross-entropy loss:

$$\mathcal{L}_{\text{CE}} = -\sum_{i=1}^{N} \sum_{c=1}^{C} \boldsymbol{y}_{i,c} \log \hat{\boldsymbol{y}}_{i,c} \tag{7}$$

where $\mathcal{L}_{\text{CE}}$ is the cross-entropy loss on a training set with $N$ samples and $C$ classes. The true labels are represented as $\boldsymbol{y}$ and the predicted labels are represented as $\hat{\boldsymbol{y}}$. For sample $i$, $\boldsymbol{y}_i$ and $\hat{\boldsymbol{y}}_i$ are $C$-dimensional vectors indexed by $c$. $\boldsymbol{y}_i$ is a one-hot vector and $\hat{\boldsymbol{y}}_i$ is a vector of predicted probabilities. A regularization term penalizing large weights is often included in the overall loss, though this term is not relevant to our discussion in this section.

**Generative Classifiers.** Probabilistic generative classifiers take an indirect approach to modelling $P(y|\boldsymbol{x})$. Namely, generative classifiers model the joint distribution $P(\boldsymbol{x}, y)$, factorized as $P(\boldsymbol{x}|y)P(y)$, then use Bayes' rule for classification:

$$P(y|\boldsymbol{x}) = \frac{P(\boldsymbol{x}|y)P(y)}{P(\boldsymbol{x})} \tag{8}$$

When classifying a single sample, the denominator $P(\boldsymbol{x})$ is a constant and can be ignored. In the numerator, $P(y)$ can be modelled by simply counting the number of instances in a given class. Therefore, the key challenge is modelling $P(\boldsymbol{x}|y)$. This can be accomplished by creating a separate generative model for each class. These models can be very simple, such as Gaussian Naive Bayes, which represents each feature's distribution for a given class as a univariate Gaussian. Alternatively, more complex models such as variational autoencoders could be used to learn each class conditional distribution.

## 4.2 Naive Approaches

**Approach #1: Single Discriminative Model for All Tasks.** We start by discussing the use of a single, discriminatively-trained model in the CIL setting. For example, consider a neural network classifier trained with cross-entropy loss. Before learning the first task, the network is instantiated with $|\mathbb{Y}_1|$ neurons in its output layer, corresponding to the number of classes in the first task. The network is trained until convergence on the first task's data $\mathcal{D}_1$. Next, $|\mathbb{Y}_2|$ additional neurons with randomly-initialized weights are added to the network's output layer prior to learning the second task. The network is trained to convergence on the second task's data $\mathcal{D}_2$, and the process repeats for the remaining tasks.

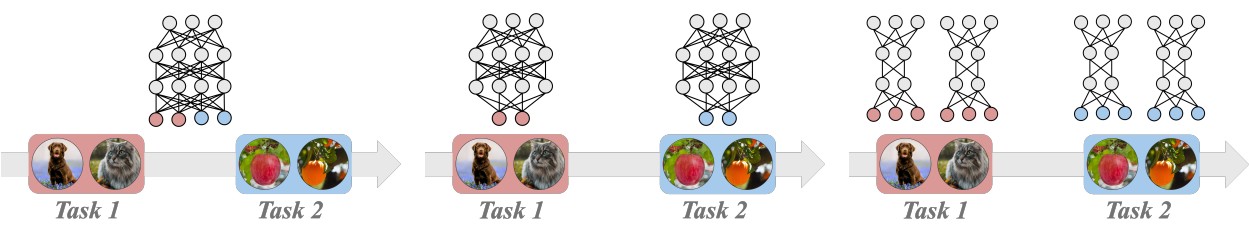

| (a) Single Discriminative | (b) Task-Specific Discriminative | (c) Class-Specific Generative |

Figure 1: Illustration of naive approaches to CIL. Two classification tasks are shown: dog versus cat and apple versus orange, shown in light red and light blue boxes respectively. Stylized neural network implementations for each approach are illustrated, with output node color corresponding to task. Discriminative approaches include one output node for each class, used to predict $P(y|\boldsymbol{x})$. The generative approach includes three output nodes for each class, illustrating a variational autoencoder which attempts to reconstruct the input. Importance sampling can then be used to predict $P(\boldsymbol{x}|y)$ – see the implementation described in Van De Ven et al. (2021) for details. Task sequence illustration inspired by Zhou et al. (2024). Best viewed in color.

This approach will almost certainly result in severe forgetting, as it does not include any mechanism to maintain knowledge learned on previous tasks. In other words, *this approach highly favors plasticity over stability*. In addition, *it is unlikely that across-task discrimination will be learned*, given that the model only performs discriminative learning within each task. Finally, *partial knowledge transfer is realized*. When learning task $k + 1$, the model is initialized with knowledge from task $k$, thereby allowing forward transfer between consecutive tasks.

The memory cost and inference compute required for a single discriminative model are both approximately constant. Here, we assume that the number of output layer weights is small relative to the model's total number of weights. The required training compute scales linearly with the number of tasks.[9]

**Approach #2: Task-Specific Discriminative Models.** Instead of using a single discriminative model for all tasks, separate discriminative models could be used for each specific task. For example, a neural network with $|\mathbb{Y}_1|$ output layer neurons could be trained on $\mathcal{D}_1$, then a separate, randomly-initialized network with $|\mathbb{Y}_2|$ output layer neurons could be trained on $\mathcal{D}_2$, and so on.

When using this approach, *both stability and plasticity are achieved* – an entirely new model is learned for each task (plasticity), and models do not interfere with one another (stability). However, *it is still unlikely that across-task discrimination is learned* – as previously discussed, learning within-task discrimination is typically inadequate for performing across-task discrimination. Lastly, *knowledge transfer will not be realized*, since task-specific models are completely separate and randomly-initialized.

The memory, training compute, and inference compute all scale linearly with the number of tasks in this context, as new models must be allocated and trained for each task. As task identification is not provided at inference, all models must be run when classifying test data.

**Approach #3: Class-Specific Generative Models.** Generative models serve as an alternative naive solution to CIL. For example, consider training a separate variational autoencoder (VAE) to model the conditional probability distribution $P(\boldsymbol{x}|y_k)$ for each class $k$. When learning the first task, $|\mathbb{Y}_1|$ VAEs are each trained in parallel on class-specific subsets of $T_1$. When learning the second task, $|\mathbb{Y}_2|$ additional VAEs are each trained in parallel on class-specific subsets of $T_2$. This process repeats for all tasks.

In this approach, *stability and plasticity are both achieved* as additional model(s) are allocated for each task. In addition, *across-task discrimination is learned*. This is because generative models are learned at the class-level rather than at the task-level, therefore class-to-task assignment is irrelevant. Still, *knowledge transfer remains unrealized* since class-specific models are learned independently.

Given that class-specific models are trained, the required memory, training compute, and inference compute all scale linearly with the number of classes. At inference, all models are run and Bayes' rule (more precisely, the numerator of Eq. 8) is used to make predictions.

---

[9]In practice, training compute may scale sublinearly with the number of tasks due to forward knowledge transfer, e.g., learning tasks $1, \ldots, i$ may improve the convergence rate when learning task $i + 1$. In general, without any assumptions on the task structure, training compute is $\mathcal{O}(T)$.

Interestingly, note that learning a generative classifier in the CIL setting is essentially the same as learning a generative classifier in the non-continual setting, as discussed in Van De Ven et al. (2021). In other words, training a generative classifier on a set of classes in the CIL setting will result in the exact same model as training the classifier in the non-continual setting. This is because generative classifiers are trained at the class-level, and do not require simultaneous access to data from multiple classes in order to learn (unlike discriminative classifiers). As a result, the predictive performance of generative classifiers is unaffected by both class-to-task assignment and task order.

Table 2: Key challenges addressed by naive (i.e., non-continual) approaches to CIL. A red x-mark indicates that a given challenge is unaddressed, while a green check mark indicates that a given challenge is addressed. An asterisk indicates that a challenge is partially addressed.

| | Stability | Plasticity | Across-Task Discrimination | Knowledge Transfer |
|---|---|---|---|---|
| Single Discriminative Model | ✘ | ✓ | ✘ | ✓* |
| Task-Specific Discriminative Models | ✓ | ✓ | ✘ | ✘ |
| Class-Specific Generative Models | ✓ | ✓ | ✓ | ✘ |

Table 3: Memory and compute requirements for naive (i.e., non-continual) approaches to CIL. The total number of tasks to be learned is $T$, while the total number of classes to be learned is $C$. Note that $C = |\mathbb{Y}_{1:T}|$.

| | Memory | Training Compute | Inference Compute |
|---|---|---|---|
| Single Discriminative Model | $\mathcal{O}(1)$ | $\mathcal{O}(T)$ | $\mathcal{O}(1)$ |
| Task-Specific Discriminative Models | $\mathcal{O}(T)$ | $\mathcal{O}(T)$ | $\mathcal{O}(T)$ |
| Class-Specific Generative Models | $\mathcal{O}(C)$ | $\mathcal{O}(C)$ | $\mathcal{O}(C)$ |

# 5 Existing Work

The observations in Section 4 suggest natural solutions to the CIL problem. Each of the three naive approaches serve as a starting point when developing CIL solutions, with modifications necessary to address the key challenges described in Section 3 under the resource constraints described in Section 2. In this section, we provide a high-level, non-comprehensive overview of existing work on CIL within this framing, covering the main ideas proposed in work published prior to September 2024. We describe the major categories of approaches and outline the key design choices for each category. An illustration these categories is shown in Figure 2.

### 5.1 Modifying a Single Discriminative Model: Replay and Regularization

To succeed in the CIL setting, a single discriminative model must be supplemented with mechanisms to avoid forgetting (i.e., maintain stability) and learn across-task discrimination. The majority of work on the CIL setting focuses on this direction (Zhou et al., 2024). Here, we discuss two categories of approaches – replay and regularization – used to supplement discriminative models in the CIL setting.

**Replay.** This set of approaches allows models to revisit previous task data when learning new tasks. A relatively small subset of previous task data is stored in a replay buffer. When new tasks arrive, the model is trained jointly on the new task data as well as the old replay data. Because the loss (Eq. 7) is optimized over samples from both the current task and previous tasks, forgetting is mitigated and across-task discrimination is learned. The data in the replay buffer is often referred to as class exemplars, and replay is sometimes called rehearsal.[10]

In general, replay-based methods are designed with the goal of limiting the size of the replay buffer while maintaining strong performance on previous tasks. The design of such methods requires answers to three key questions: (1) Which samples should be stored? (2) How should these samples be stored? (3) How should these samples be replayed during training?

To address question (1), previous work has proposed the use of herding algorithms (Welling, 2009) to select "prototypical" samples (Rebuffi et al., 2017), maximizing sample diversity with respect to model gradient updates (Aljundi et al., 2019b), and selecting difficult-to-classify samples based on prediction entropy (Chaudhry et al., 2018b).

Solutions proposed to address question (2) include storing learned representations (Iscen et al., 2020) or compressed features (Zhao et al., 2021) rather than raw features, or learning a generative model in order to create synthetic data for replay (Shin et al., 2017; Hu et al., 2018; Kemker & Kanan, 2018) (sometimes called generative replay or pseudo-rehearsal). While generative replay approaches require learning a generative model, this model is used to generate data for a downstream discriminative model, hence their grouping with other discriminative models.

To answer question (3), most approaches (Rebuffi et al., 2017; Chaudhry et al., 2018b; Aljundi et al., 2019b; Zhao et al., 2021; Iscen et al., 2020; Shin et al., 2017; Hu et al., 2018; Kemker & Kanan, 2018) use a straightforward application of replay – for each task, the model minimizes the loss over both the current task data and the previous task data from the replay buffer. Alternatively, some approaches (Lopez-Paz & Ranzato, 2017; Wang et al., 2021; Zeng et al., 2019; Tang et al., 2021) frame learning the current task as a constrained optimization problem in which model parameters may only be updated such that the loss on the replay data does not increase (or increases only by a small amount).

Replay-based approaches have been observed to be biased towards the recently learned classes. This bias is due to data imbalance, as there is typically much more data available in the current task as compared to the previous tasks. Several methods have been proposed to mitigate this bias. These methods include the addition of a simple rectification layer used to adjust predictions (Wu et al., 2019), making use of statistical information from previous classes for de-biasing (Belouadah & Popescu, 2019), and compensating for bias at the gradient level during optimization (Guo et al., 2023).

**Regularization.** This second set of techniques attempts to retain previously learned knowledge using a regularization term in the loss function. This regularization term penalizes changes in model behavior with respect to old tasks. Note that this penalty is used to mitigate forgetting, though it is typically insufficient for learning across-task discrimination (Lesort et al., 2019). Regularization approaches fall into two subcategories: parameter-based and data-based regularization (De Lange et al., 2021).

Parameter-based regularization approaches attempt to retain parameter values which are important for strong performance on previously-learned tasks. These parameters should not change when learning new tasks, otherwise forgetting may occur. Intuitively, only the unimportant parameters for tasks $1, \ldots, k$ may be modified when learning task $k + 1$. Many approaches in this subcategory use Fisher Information as an

---

[10]More precisely, rehearsal refers to methods which use replay data for retraining (Bagus & Gepperth, 2021). Some methods instead use replay data to generate constraints while learning new tasks.

importance measure (Kirkpatrick et al., 2017; Lee et al., 2017; Chaudhry et al., 2018a; Yang et al., 2019; 2023b). Some approaches compute parameter importance after tasks have been learned (Kirkpatrick et al., 2017) while others compute importance during learning (Zenke et al., 2017; Chaudhry et al., 2018a).

Data-based regularization methods typically leverage knowledge distillation to mitigate forgetting. Knowledge distillation is traditionally used to transfer knowledge from a larger teacher model to a smaller student model in an attempt to replicate the teacher's performance using the student (Hinton et al., 2015). In continual learning, the model(s) trained on previous tasks serve as the teacher, and the model trained on the current task serves as the student. The student model attempts to learn the current task while replicating the performance of the teacher on previous tasks, accomplished via an additional distillation term in the loss function. Distillation may be performed at the level of the logits – the final layer of the neural network (Li & Hoiem, 2017; Rebuffi et al., 2017; Zhang et al., 2020b; Hou et al., 2018; Smith et al., 2021; Lee et al., 2019; Zhou et al., 2021) – or at the level of the learned representation – an intermediate layer of the network (Simon et al., 2021; Lu et al., 2022; Kang et al., 2022; Hu et al., 2021b). Some approaches (Rebuffi et al., 2017; Zhao et al., 2020; Lee et al., 2019) combine data-based regularization with replay to improve distillation performance using a set of exemplars from previous tasks.

Similar to replay, regularization-based approaches tend to be biased toward the current task data. In addition, regularization-based approaches attempt to maintain separability across classes *within* each previously learned task, but are typically unable to learn separability *across* tasks. A number of methods have been proposed in an effort to address these issues. (Zhao et al., 2020) and (Hou et al., 2019) ensure that weights have comparable magnitudes across both old and new classes to achieve fair classification, (Ahn et al., 2021) separates softmax activations in the output layer to avoid task-recency bias, and (Castro et al., 2018) uses balanced fine-tuning to stabilize predictions.

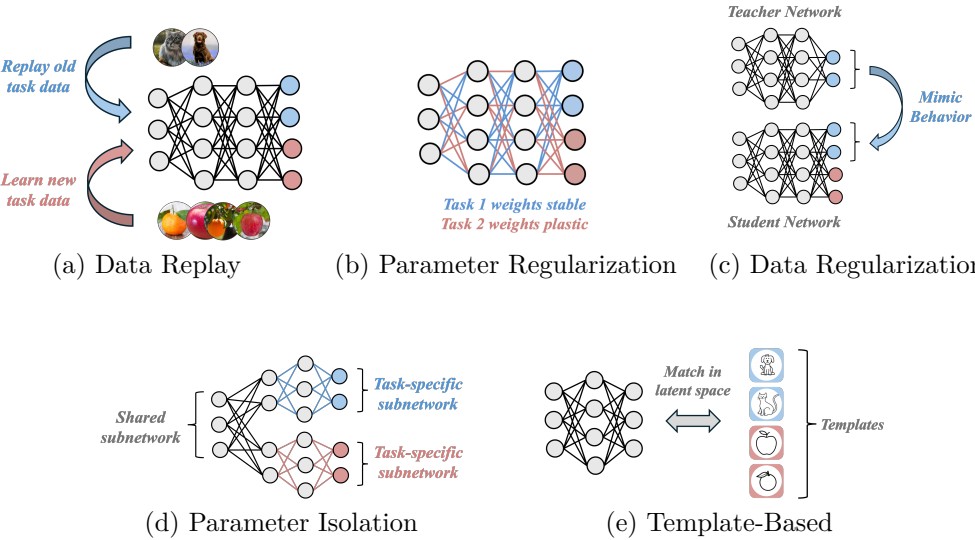

(a) Data Replay  (b) Parameter Regularization  (c) Data Regularization

(d) Parameter Isolation  (e) Template-Based

Figure 2: Illustration of existing work on CIL. Each subfigure depicts a stylized neural network, representing the high-level idea behind a given category of approaches. "Template-based" approaches include generative classifiers and hybrid models. These approaches match input samples to learned templates – class prototypes or class-conditional distributions – in order to perform classification. Best viewed in color.

## 5.2 Modifying Task-Specific Discriminative Models: Parameter Isolation

Both stability and plasticity can be achieved by allocating separate discriminative models for each task. However, this approach lacks mechanisms to learn across-task discrimination and exploit knowledge transfer. In addition, learning completely separate models for each task may be inefficient with respect to memory and compute. Parameter isolation methods address these challenges.

**Parameter Isolation.** This family of methods allocates sets of discriminatively-trained parameters (i.e., submodels) specific to each task that may not be modified when learning other tasks. Parameter isolation is a simple way to avoid forgetting – if previously learned knowledge is never modified, then it will never be overwritten when learning continually. However, parameter isolation methods introduce a new problem: at inference, they require an additional task-selector model – used to predict the task label – in order to choose the relevant task-specific submodel. In other words, most parameter isolation approaches decompose the CIL problem into task prediction (TP) and within-task prediction (WP) (Kim et al., 2022). Approaches to TP include using model prediction entropy to infer task label (Wortsman et al., 2020), creating a separate network as a dedicated task predictor (Abati et al., 2020), and utilizing out-of-distribution detection models (Kim et al., 2022).

Parameter isolation methods can be divided into two subcategories – fixed architecture and dynamic architecture – based on their approach to WP. Fixed architecture methods start with a fixed-size network and allocate task-specific subnetworks during training. Approaches to subnetwork allocation include the use of weight masks (Serra et al., 2018; Wortsman et al., 2020), genetic algorithms (Fernando et al., 2017), and pruning (Mallya & Lazebnik, 2018). Alternatively, dynamic architecture methods do not have a fixed size, and expand as necessary in order to learn new tasks. These approaches are concerned with efficiently expanding model size as new tasks arrive in order to minimize memory cost while maintaining strong performance. Various techniques have been proposed to accomplish this goal, including dynamically creating and deleting neurons (Yoon et al., 2018), framing the problem as reinforcement learning (Xu & Zhu, 2018) or neural architecture search (Li et al., 2019), using compression (Schwarz et al., 2018) or distillation (Wang et al., 2022a) to lower the model's memory footprint, and focusing expansion on specific components to better transfer knowledge across tasks (Zhou et al., 2023). Interestingly, Kim & Han (2023) finds that many CIL methods heavily favor stability over plasticity, and proposes holding earlier layers stable while expanding later layers in order to strike a fairer balance between old and new tasks.

### 5.3 Modifying Class-Specific Generative Models: Generative Classifiers, Hybrids, and Pretraining

Recall that generative classifiers are a natural fit for CIL – they achieve both stability and plasticity, and learn across-task discrimination without requiring a task-prediction model. However, traditional generative classifiers suffer from poor memory and compute scaling and lack mechanisms to leverage knowledge transfer.

In addition, the predictive performance of generative classifiers is typically inferior to that of discriminative classifiers in the non-continual setting. This statement is supported by a wealth of theoretical and empirical evidence, see Vapnik (1998) and Ng & Jordan (2001). At a high level, this is because generative classifiers solve a more difficult problem – intermediately modelling $P(\boldsymbol{x}|y)$ – while discriminative classifiers solve an easier problem – directly modelling $P(y|\boldsymbol{x})$. Generative classifiers and hybrid generative-discriminative models designed for CIL attempt to remedy these issues through modifications which improve resource efficiency, leverage knowledge transfer, and boost predictive performance.

**Generative Classifiers.** In addition to the large body of work on discriminative classifiers for CIL, there exists a smaller body of work focused on generative classifiers. Van De Ven et al. (2021) proposes generative classifiers for the CIL setting and provides results with a class-specific variational autoencoders (VAEs) as a proof-of-concept. Other approaches also utilize VAEs (Ye & Bors, 2021a; 2023) as well as generative adversarial networks (GANs) (Ye & Bors, 2020; 2021b) but focus on ways to efficiently expand the architecture and exploit knowledge transfer between submodels.

**Hybrid Models.** Some approaches cannot be cleanly categorized as generative or discriminative. These approaches include prototype-based classifiers, one-class classifiers, and hybrid generative-discriminative classifiers. Many of these approaches retain the desirable properties of generative classifiers while avoiding the drawbacks of strictly discriminative classifiers.

Prototype-based classifiers (Zhu et al., 2021b; De Lange & Tuytelaars, 2021) learn prototypical representations for each class and use nearest neighbor matching for classification. Similar to Naive Bayes, prototype-based classifiers are natural continual learners, as new prototypes can be learned as new classes are introduced (Grossberg, 2020; Ashtekar & Honavar, 2023). One-class classifiers (Hu et al., 2021a; Sun et al., 2023) learn independent models for each class. These models are optimized to predict high values for in-class data –

similar to conditional generative models – though one-class classifiers use different objective functions (Hu et al., 2020).

Hybrid generative-discriminative classifiers (Kirichenko et al., 2021), combine the lack-of-forgetting of generative models with the strong classification ability of discriminative models. Generative techniques can be used to model features "holistically" (Hu et al., 2021a), while discriminative techniques can be used to enhance separation between classes. More generally, *prospective modeling* (Tian et al., 2024) may be necessary to address the across-task discrimination challenge: when learning class $i$, it is unclear which features will be discriminative against a future class $j$. Generative of pseudo-generative techniques can be used to model both the features which are discriminative across the previously learned classes and the features which may be discriminative with respect to future classes. Other approaches in this direction include allocating "virtual" or "reserved" space in the learned representation as a proxy for future classes (Zhou et al., 2022a; Song et al., 2023; Zhou et al., 2022b).

**Pretraining.** Another category of approaches avoids continual representation learning through the use of pretrained models. Wang et al. (2022d) and Wang et al. (2022c) introduce the idea of *learning to prompt* for continual learning. These approaches learn to instruct pretrained transformers to perform tasks in the continual setting. Other approaches learn simple generative classifiers on features produced by fixed pretrained networks (Yang et al., 2023a; McDonnell et al., 2023). For example, Yang et al. (2023a) learns a Naive Bayes classifier atop features generated by a network trained on ImageNet in order to perform downstream image classification tasks.

Pretraining is particularly relevant in the few-shot setting (FSCIL), in which only a small amount of data is available when learning each class. Many approaches to FSCIL focus on learning in the later network layers with a fixed, pretrained representation. Shi et al. (2021) observes that FSCIL performance improves when starting from a pretrained network which has converged to a flat local minima in the loss landscape. In a flat local minima, small parameter changes will leave performance relatively unaffected, thus mitigating forgetting while learning new classes. Other approaches to FSCIL utilize meta-learning, or "learning how to learn", typically accomplished through bi-level optimization. Chi et al. (2022) extends the classic model-agnostic meta-learning (MAML) algorithm first proposed in Finn et al. (2017) for the FSCIL setting.

## 6 Dimensions of the Problem

Given the abundance of work on CIL, it is natural to ask when each approach should be used. We argue that the appropriate choice of approach depends on the specific dimensions of a given CIL problem. In this section, we outline five such dimensions: *resource constraints, task size, task similarity, task order,* and *class-to-task assignment*. We explain each dimension, describe how it influences the difficulty of the CIL problem, and discuss how it is addressed by existing approaches.

**Resource Constraints.** Resource constraints are arguably the most important dimension of the CIL problem: constraints on available memory and compute define the set of potential solutions. Generally speaking, stricter resource constraints make the problem harder, while looser constraints make the problem easier. For example, given no resource constraints, CIL can be trivially solved by retraining on *all* learned data $\mathcal{D}_{1:k}$ at every task $k$. This approach matches the predictive performance in the non-continual setting and renders continual learning methods unnecessary. On the other hand, if an application requires tight memory and/or compute budgets, then more care is required to design bespoke continual learning solutions. These solutions should be designed with the goal of approaching – and ideally, matching – the predictive performance of a model learned in the non-continual setting in which data from all the tasks is available.

Recent work indicates that *replay* may be the most promising deep learning solution to CIL under strict resource constraints. Specifically, replay-based approaches have been shown to outperform regularization-based and parameter isolation-based approaches when given comparable budgets for memory (Zhou et al., 2024) and compute (Prabhu et al., 2023a). Notably, Harun et al. (2023b) proposes SIESTA – a wake-sleep algorithm utilizing replay – which matches the performance of a non-continual learner on the ImageNet-1K dataset under a tight compute budget. Other promising approaches for resource-efficient CIL with neural

networks include pretraining (McDonnell et al., 2023), layer-specific plasticity (Zhou et al., 2023), dynamic weight/gradient-masking (Wang et al., 2022b), and sharpness-aware minimization (Ren & Honavar, 2024).

**Task Size.** Performing CIL with larger tasks (tasks containing more classes) is typically easier than performing CIL with smaller tasks. More precisely: for a chosen dataset, CIL methods tend to perform better given a few large tasks as compared to many small tasks – see Masana et al. (2020) and Zhou et al. (2024) for empirical evidence on the split CIFAR-100 and ImageNet-1K datasets. Consider the extreme cases: (i) there is only one task containing all classes and (ii) each task contains a single class. Case (i) is equivalent to the non-continual setting, so continual learning is unnecessary. Case (ii) is a version of CIL which does not include within-task discrimination, as there is only one class-per-task. In other words, the CIL problem in case (ii) is entirely across-task discrimination. For existing methods, learning across-task discrimination tends to be more challenging than learning within-task discrimination (Soutif-Cormerais et al., 2021; Guo et al., 2023), therefore case (ii) can be categorized as the "most difficult" version of CIL.

To understand why, consider the following. Recall that discriminative classifiers require simultaneous access to data from all classes in order to effectively learn, and that discriminative classifiers typically outperform generative classifiers. As described in Section 5, across-task discrimination can be addressed in one of two ways: with a discriminative classifier supplemented by replay, or with a generative/pseudo-generative classifier. Resource constraints restrict the amount of old data stored and/or the number of learning updates completed during replay, thus limiting the performance of discriminative approaches. On the other hand, generative/pseudo-generative techniques offer lower performance than their discriminative counterparts. Either way, we would expect the performance on across-task discrimination to be lower than the performance on within-task discrimination.

**Task Similarity.** There are various ways to define similarity between classification tasks. These definitions can be categorized as either model-dependent – based on a particularly model architecture or training process – or model-agnostic – independent of the type of model or training procedure used. Model-dependent similarity can be defined using knowledge transfer (see Eq. 5 and Eq. 6), learned parameter values (Lee et al., 2021), or the distance between pretrained task embeddings (Achille et al., 2019). Model-agnostic similarity can be defined using optimal transport (Alvarez-Melis & Fusi, 2020; Liu et al., 2025) and/or conditional entropy (Tran et al., 2019; Tan et al., 2021).

CIL approaches such as replay exploit task similarity through parameter initialization: when learning task $k$, the model is initialized with the knowledge from tasks $1, \ldots, k-1$. This can help reduce the number of training iterations when learning task $k$, thus improving computational efficiency. Approaches such as replay, regularization, partial parameter isolation, and generative modelling exploit task similarity through parameter sharing. Sharing a subset of parameters across tasks can reduce computational and memory costs. In addition, both parameter initialization and parameter sharing have the potential to improve predictive performance, as evidenced by the large body of work on transfer learning (Zhuang et al., 2020).

While task similarity can be exploited to improve learning, attempting to learn similar tasks presents additional challenges. Ramasesh et al. (2021) and Nguyen et al. (2019) empirically show that increasing task similarity often increases the severity of catastrophic forgetting. Through theoretical analyses of continual linear regression, Evron et al. (2022) and Lin et al. (2023) find that forgetting is greatest when learning tasks with *intermediate* similarity. Namely, regression tasks with common features but misaligned weights[11] are the most prone to forgetting. Designing CIL solutions which leverage task similarity without introducing across-task interference remains an interesting problem.

**Task Order.** The performance of most CIL methods depends on the order in which tasks are presented. For example, Masana et al. (2020) evaluate several popular CIL methods across various class (and therefore task) orderings of the CIFAR-100 dataset and report wide variations in performance. Further, the method which performs best also depends on the ordering! This indicates the importance of evaluating CIL methods over multiple task orderings and class-to-task assignments.

---

[11]Formally, let the optimal linear regression weights for task 1 and task 2 be $\boldsymbol{w}_1^*$ and $\boldsymbol{w}_2^*$. The "misaligned weights" condition corresponds to $\langle \boldsymbol{w}_1^*, \boldsymbol{w}_2^* \rangle < 0$, as described in Lin et al. (2023)

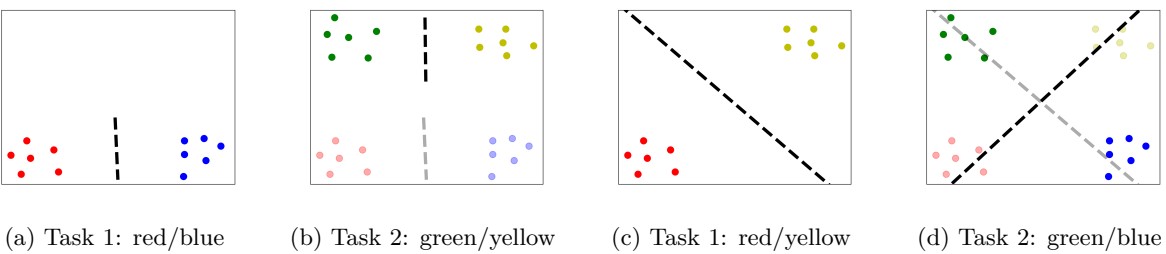

(a) Task 1: red/blue      (b) Task 2: green/yellow      (c) Task 1: red/yellow      (d) Task 2: green/blue

Figure 3: Example illustrating the importance of class-to-task assignment. For a given set of classes, (a) and (b) illustrate one class-to-task assignment, while (c) and (d) illustrate another. Data samples are represented as shaded circles, with color indicating class label. Learned decision boundaries are represented with black dotted lines. Figure inspired by Lesort et al. (2019). Best viewed in color.

*Curriculum learning* is a subfield of machine learning which attempts to build training schedules in order to improve learning efficiency and performance (Bengio et al., 2009). While continual learning is concerned with efficiently learning a set of sequential tasks, curriculum learning is instead concerned with designing a task sequence which can be efficiently learned. For a given dataset, a curriculum is an ordering[12] of the dataset's samples to be learned during training. Curricula include learning "easy" samples before "hard" samples, learning "hard" samples before "easy" samples (anti-curriculum), and emphasizing sample diversity early in training (Xin Wang, 2022). Note that curricula are typically designed at the sample-level, though Pentina et al. (2015) extends these ideas to the task-level.

Most work on CIL assumes that task order is fixed (i.e., not chosen by the model) and that models have no knowledge of future tasks prior to learning. Several papers (Ruvolo & Eaton, 2013; Yang & Li, 2021; Bell & Lawrence, 2022; Mantione-Holmes et al., 2023) relax this assumption and attempt to find the task/class order which maximizes overall classification performance. Bell & Lawrence (2022) propose ordering tasks such that the distance between optimal task-specific parameters is small. However, their results are counterintuitive: performance on computer vision benchmarks is strongest when learning tasks in an order which results in large parameter changes. Mantione-Holmes et al. (2023) report similar results. Inspired by psychology, they find that an interleaved task order (learning dissimilar tasks adjacently) improves performance on NLP classification problems. Lin et al. (2023) provide evidence supporting this observation through a theoretical analysis of linear models.

When tasks are interleaved, the model is exposed to a diverse set of classes all throughout training. This may better approximate the non-continual setting in which data from all classes is available, thus explaining the relatively strong performance on interleaved orderings. Interestingly, Yang & Li (2021) come to a different conclusion: learning similar tasks adjacently results in the strongest overall performance on image classification tasks. These findings may be application-specific, or the differences may be reconciled by considering the various experimental setups and definitions of task similarity used across studies.

**Class-to-Task Assignment.** The way in which classes are assigned to tasks affects the performance of most CIL methods. Klasson et al. (2023) provides empirical evidence of this phenomenon across various replay-based methods on the split MNIST series and CIFAR-10 datasets with randomized class-to-task assignments. As with task order, this dimension of the problem has been observed to affect CIL performance, though this effect is not well understood.

Figure 3 illustrates the importance of class-to-task assignment on a simple CIL problem with linear classifiers learned on each task. Here, we are given a set of four classes, and we consider two possible class-to-task assignments. The first assignment – shown in subfigures 3a and 3b – results in roughly the same within-task decision boundaries for both tasks, though across-task discrimination must be learned separately. The second assignment – shown in subfigures 3c and 3d – results in nearly orthogonal within-task decision boundaries, and across-task discrimination is implicitly learned. In other words, the CIL problem in the second assignment can be solved by task-specific discriminative models alone.[13]

---

[12]Curricula can also be defined using instance selection or sample reweighting – see Xin Wang (2022) for details

[13]For example, two separate logistic regression models – one for Task 1 and one for Task 2 – could be used, with predictions made based on the maximum logit activation across both models

We hypothesize that the performance of CIL models will be stronger when classes are assigned to tasks such that the similarity of classes *within-task* is greater than the similarity of classes *across-task*. Here, class similarity can be quantified using the performance of a given classifier – for a set of classes, if classifier performance is low, then the classes can be said to be similar, and if classifier performance is high, then the classes can be said to be dissimilar. In other words, we hypothesize that CIL performance will be higher if the across-task discrimination subproblem is easier relative to the within-task discrimination subproblem. We expect this hypothesis to be true given that across-task discrimination is a larger part of the overall CIL problem as compared to within-task discrimination, as described in Section 3. It may be reasonable to expect this condition to hold in real-world applications. This is because new tasks are likely to result from focused data collection and/or from temporally correlated changes in the world.

## 7 Conclusion and Future Work

At this point, we take a step back and reflect on our progress. We began by reexamining the CIL problem definition, arguing that common framings neglect important considerations such as computational cost. Next, we explored the factors which make CIL unique and decomposed the problem into three key challenges. With these challenges in mind, we analyzed non-continual approaches to the problem. The shortcomings of these approaches served to motivate and contextualize existing work on CIL. After summarizing existing work, we investigated various dimensions of the problem in order to better understand specific problem configurations and their corresponding solutions. Now, we look ahead and suggest potential directions for future work.

### 7.1 Future Work

**Developing computationally efficient solutions.** In Section 2, we discuss why the CIL problem should be defined based on resource constraints rather than data availability. Note that the arguments supporting the resource-constrained problem statement extend beyond the CIL setting. We argue that other continual learning settings such as task incremental learning (TIL), domain incremental learning (DIL), continual reinforcement learning, etc. should also be defined with respect to resource constraints.

As analyzed in Verwimp et al. (2024), the majority of continual learning papers at recent conferences[14] highly constrain memory, but do not constrain computational cost. These constraints may be counterproductive, as "memory is cheap, but compute is expensive" nowadays given contemporary hardware and cloud computing platforms (Lomonaco & Carta, 2023). Instead, it may be more appropriate to constrain or optimize for low computational costs under large (or even unlimited) memory budgets.

The discussion in Section 2.4 on a potential real-world CIL system provides preliminary ideas in this direction: matching-based or other locally-weighted classifiers could be learned atop a continually evolving representation, updated via replay and/or knowledge distillation. While continual representation learning has been well-studied under the "unavailability of previous data" problem statement (Section 2.2.1), relatively little work has addressed this problem in the context of computational efficiency beyond Li et al. (2022); Harun et al. (2023b), and Prabhu et al. (2023a).

**Making assumptions regarding the data-generating process.** Most work on CIL assumes that models have no knowledge of future tasks prior to learning, as mentioned in Section 6. We argue that this assumption is unnecessarily restrictive and unlikely to hold in practice. To understand why, consider the manifold hypothesis, which states that real-world, high-dimensional datasets tend to be concentrated along low-dimensional manifolds (Narayanan & Mitter, 2010). For example, real-world images tend to be highly structured, and are unlikely to resemble a random collection of pixels. In an application of CIL for image classification, all classes are likely to share some common structure. If this structure is known apriori, learning may be made easier.[15]

---

[14]ECCV '22, CVPR '23, NeurIPS '22, and ICML '23

[15]Interestingly, many CIL approaches for image classification do make use of this common structure – for example, by sharing convolutional layers across tasks. However, the assumption that classes or tasks have a common structure is often not explicitly stated or formalized.

As another example, consider a continual learning application leveraging physics-informed neural networks (Howard et al., 2024). The laws of physics remain constant as new tasks are presented (this fact is known before any tasks are learned). In this example, one aspect of the data-generating process is fixed, while other aspects may vary. For a given application, it may be possible to explicitly specify which aspects of the data-generating process are changing. Such specifications could be used to build better inductive biases, thus improving learning efficiency and performance. Doing so may also circumvent some of the negative results regarding the difficulty of continual learning (Knoblauch et al., 2020).

**Science versus engineering in continual learning.** Looking back at the past eight years of work on continual learning, we observe the following high-level trend. Early work (2016 - 2019) introduced new categories of approaches to the problem: Elastic Weight Consolidation (Kirkpatrick et al., 2017) introduced parameter regularization, Learning without Forgetting (Li & Hoiem, 2017) introduced knowledge distillation, Progressive Neural Networks (Rusu et al., 2016) introduced parameter isolation, and so on. More recent work has focused on refining the best-performing solutions and understanding why they work. This trend is analogous to the exploration-exploitation trade-off in reinforcement learning – early work explored the solution space, while recent work exploits the most successful solutions.

Work on continual learning can be very loosely divided into two categories: science and engineering. Scientific work includes proposing novel algorithms (often inspired by neuroscience), understanding why various techniques succeed or fail, and proving theoretical results regarding complexity, performance, convergence, etc. Engineering work is concerned with developing efficient and effective solutions for specific applications. While there is a great need for further engineering advances in continual learning, there is an equal, if not greater, need for elucidating the theoretical underpinnings of continual learning. Future engineering work will likely focus on specific, real-world applications, moving beyond the current general-purpose, somewhat contrived benchmarks such as the split versions of the MNIST, CIFAR-100, and ImageNet-1K datasets. We expect that improvements in practical continual learning applications will lead to an increased adoption of continual learning in industry, further driving progress.

### 7.2 Actionable Insights

We conclude with several high-level actionable insights for future CIL research.

1. **Address the resource-constrained setting.** As discussed in Section 2, framing CIL (and continual learning more generally) based on the unavailability of previous data may lead to undesirable outcomes. This framing is imprecise, may not adequately address security or privacy concerns, and permits inefficient solutions. We argue that the resource-constrained setting does not suffer from these drawbacks and is more appropriate for real-world applications, particularly with its inclusion of compute constraints.

2. **Ensure that plasticity is available for future tasks.** Much of the work on continual learning focuses on avoiding catastrophic forgetting (i.e., maintaining stability). However, the focus on avoiding forgetting often neglects plasticity, limiting performance on future tasks.

3. **Across-task discrimination dominates the CIL problem – build solutions accordingly.** At inference, the notion of separate tasks in CIL is somewhat artificial: the one "true" task is to accurately make predictions across all previously learned classes. Therefore, across-task discrimination is almost always a larger part of the problem as compared to within-task discrimination.

4. **Dimensions of the CIL problem are poorly understood and ripe for future work.** Open questions remain concerning the impact of task similarity, class/task order, and class-to-task assignment on CIL performance. Answering these questions could lead to better CIL algorithms as well as a deeper understanding of the broader continual learning problem.

5. **Use (reasonable) assumptions about your data to make learning easier.** It may be possible to make assumptions regarding the data generating process in specific continual learning applications (i.e., physics-informed learning). Doing so could improve learning efficiency and performance.

## Acknowledgments

This work was supported in part by grants from the National Science Foundation (2226025) and the National Center for Advancing Translational Sciences, and the National Institutes of Health (UL1 TR002014) and by the Center for Artificial Intelligence Foundations and Scientific Applications and the Institute for Computational and Data Sciences at Pennsylvania State University.

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
