# OpenReview forum: "Class Incremental Learning from First Principles: A Review"
_TMLR — Accepted by TMLR_

### Review · Reviewer_EWou · 2024-10-26

**Summary Of Contributions:**

This paper provides an alternative perspective to existing work on CIL. Considering resource constraints revisiting the problem definition and describing its unique challenges. A principled way of developing solutions to the CIL problem is provided by analysing discontinuous methods and providing an overview of existing work. Five specific dimensions of the CIL problem are then analysed and how existing work deals with various problem configurations is described. Recommendations for future work are summarised at the end of the article, arguing that future solutions can and should make assumptions regarding the data generation process.

**Audience:**

Yes

**Broader Impact Concerns:**

1)	The CIL framework proposed in the article may rely on storing some of the old data or generating data for playback by generating models. However, this may lead to privacy and data security issues, especially when models are trained on tasks containing sensitive data, with an increased risk of data leakage. Models may ‘remember’ old data and inadvertently disclose sensitive information, especially when applied in domains such as healthcare or finance. It is recommended that the authors explore how to enhance privacy protection mechanisms in CIL.

2)	Class incremental learning (CIL) methods may inherit biases in existing data when applied to real-world scenarios. For example, when applied to sensitive domains such as autonomous driving or facial recognition, models may amplify racial, gender, or other social biases in the data as they incrementally learn new tasks. Discussion of these social and ethical issues is lacking in the paper, so it is recommended that content be added on how to address bias and fairness issues in CIL methods.

**Claims And Evidence:**

Yes

**Requested Changes:**

1)	It is recommended that the authors add a discussion of recent research results in the field of Class Incremental Learning (CIL) beyond 2023, especially hybrid models and new benchmarking methods. You can discuss recent review papers such as;  Class-incremental learning: A survey; A survey on few-shot class-incremental learning; and others.

2)	The paper provides a broader discussion of the challenges in class-based incremental learning, and it is recommended that this section be augmented with some more detailed analyses of sub-domains, such as the differences between task-independent incremental learning and task-related incremental learning, as well as the distinctive challenges they each face. This would allow for a more nuanced discussion and meet the needs of a diverse group of readers.

3)	The paper discusses too much about the basic ‘naive’ methods (e.g., single discriminant model, task-specific model, category-generating model), and it is suggested to simplify this part of the paper and devote more space to the in-depth analyses and comparisons of the existing state-of-the-art methods, so as to emphasise the innovativeness and unique insights of the paper.

4)	When discussing the impact of resource constraints on the CIL problem, it is recommended that this section be further refined and expanded, especially the specific implementation, comparison and analysis of experimental results for different resource constraints (e.g. computation and memory), to provide more empirical support and make this part of the discussion more robust and comprehensive.

5)	In the future work section, it is suggested that more specific research directions are proposed, especially providing actionable suggestions on how to deal with cross-task discrimination learning, knowledge transfer, and how to optimise in terms of computational efficiency. This will make the article more instructive for future research.

6)	Need to summarise previous incremental learning reviews and plot them in a table showing the differences between this review compared to them, etc.

7)	Clearly state the timeframe covered by this review, i.e. all classes of incremental learning methods discussed are research results published before a specific point in time. For example, ‘This review covers all relevant work published up to October 2024’.

**Strengths And Weaknesses:**

**Strengths:**

1)	The paper is well-organized and easy to follow.

2)	The perspective is novel, and directing attention to unexplored aspects of the CIL problem.

3)	The article redefines the problem of Class Incremental Learning (CIL), explicitly pointing out the shortcomings in the existing definition, especially in the discussion of resource constraints. This re-examination of the problem provides a clearer direction and framework for subsequent research, and helps to advance the field.

4)	The article delves into three core challenges involved in class incremental learning: stability-plasticity balance, cross-task discriminative learning, and knowledge migration. These analyses not only explain the complexity of CIL compared to traditional machine learning, but also suggest directions worth exploring for future research.

5)	The article not only summarises existing methods, but also points out their shortcomings in terms of computational efficiency, cross-task differentiation, and so on. In particular, the discussion of ‘naive’ approaches provides valuable background information for understanding the difficulties and directions of CIL development.


**Weaknesses:**

1)	Although the paper differentiates itself from other reviews of continuous learning, it covers fewer of some of the most recent advances in the field of class-based incremental learning (CIL) beyond 2023, particularly emerging hybrid methods and new benchmarking. This may limit its reference value to researchers seeking to understand the latest advances in the field.

2)	The challenges of class-incremental learning are discussed primarily at a high level. A more nuanced breakdown with a focus on specific subfields (e.g., task-agnostic versus task-aware incremental learning) could provide a more granular understanding and be beneficial to different categories of readers.

3)	The section dedicated to discussing "naive" approaches such as single discriminative models, task-specific models, and class-specific generative models, while providing historical context, takes up significant space and repeats insights that are well-known. This may not add much value to the paper’s intended audience, who may already be familiar with these basic approaches.

4)	The argument for framing CIL in terms of resource constraints instead of unavailability of past data is interesting. However, the discussion could be expanded to address specific practical implementations and comparisons, especially around the computational cost of different methods. This would make the argument more compelling and actionable for real-world applications.

5)	While the conclusion does suggest some future directions, the suggestions are somewhat broad and lacking in concrete specifics on how certain challenges (e.g., learning across-task discrimination or computational efficiency) might be addressed. This limits the practical impact and future utility of the paper.

---

> ### Author Response · Authors · 2024-10-29
> **Preliminary response to reviewer EWou**
>
> Thank you for your detailed feedback. Especially helpful were the suggestions to (i) expand the resource constraint discussion, (ii) clarify our contributions relative to other reviews/recent work, and (iii) discuss privacy-related broader impacts in CIL.
>
> We are working on the changes now, but will wait until all reviews are submitted prior to uploading a revised draft. We appreciate your help in improving the paper!

---

> ### Author Response · Authors · 2024-11-20
> **Changes in revision in response to Reviewer EWou**
>
> We have uploaded a revised version of the draft with changes listed in green text. We highlight how the changes address the reviewer's concerns below, in 1-to-1 correspondence with the "Requested Changes" provided by the review:
>
> 1. Added subsection on pretraining and few-shot CIL in Section 5
> 2. Added subsection on related problem statements (few-shot CIL, TIL, DIL, online CL) in Section 2
> 3. We appreciate this feedback, however we are unsure if other readers (particularly those less familiar with continual learning) would find parts this section redundant. For now, we will leave this section unchanged, but welcome additional feedback from the action editor.
> 4. Added potential real-world application of resource-constrained CIL in Section 2 along with discussion of potential solutions
> 5. Added brief discussion in Section 7 as well as a new subsection 7.2 on actionable insights for CIL researchers
> 6. Added Table in Section 1 summarizing existing reviews at a high-level -- we agree that this is a cleaner way to present content from the introduction -- thanks for this suggestion
> 7. Stated timeframe in Section 5
>
> Regarding broader impacts: the points brought up are pertinent, however we are unsure if a broader impacts section is required given that this work is a survey. We will defer to the recommendation of the action editor.
>
> Again, we appreciate the detailed feedback, and feel that the requested changes have improved the paper. Thank you for your help!

---

### Review · Reviewer_Ep3w · 2024-11-04

**Summary Of Contributions:**

The machine learning community has made significant strides in various applications, including computer vision and natural language processing, through the dominant approach of deep learning. However, this success is hindered by the stability-plasticity dilemma, where existing methods struggle to accommodate new data without forgetting previously learned knowledge. Continual learning has emerged as a promising solution, with a focus on alleviating catastrophic forgetting. Specifically, Class-Incremental Learning (CIL) addresses the supervised classification problem in continual learning, where new classes of data are introduced over time. This review delves into the CIL setting, examining the problem statement, existing solutions, and unexplored directions, with the ultimate goal of understanding and refining the CIL problem statement to drive future advancements.

**Audience:**

No

**Claims And Evidence:**

No

**Requested Changes:**

- To begin with, the reformulation of CIL is full of arguments that are not backed by evidence, theory, or any strong logic. The whole section needs to go. The whole paper requires a full rewrite with the existing definition of CIL established by prior works in this literature.
- For the remaining paper, there are no experimental results to compare different methods. The paper must include such results to understand the effectiveness of different methods and do the comparison.

**Strengths And Weaknesses:**

Weaknesses:
- This paper introduces/argues with terms and concepts that are not conventional and lack justifications:
    - For instance, "We argue that restrictions in the form of resource constraints are more appropriate than restrictions on the availability of previously learned data." - this is not quite valid, in my opinion. In the modern era of machine learning, where we train foundation models with ever-increasing sizes, the resources required for continual learning are not our primary concern, especially in class incremental learning settings. So, the continual learning literature motivates the problem setup as the unavailability of previously learned data for various reasons, including privacy concerns.
- The argument becomes more problematic in section 2.1.
    - If we remove the old restriction and we have all the previous data available, we do not need a specialized continual learning solution. We can simply rely on a supervised solution. The entire point of the literature and the existing methods is to make sure to learn from new data while not forgetting the old ones. Even rehearsal-based solutions that store a few samples per old class perform considerably higher than solutions that do not store previous samples.
    - Again, Critique #1 is not a significant problem. Also, the training compute time is less of a concern compared to inference compute time.
    - Critique #2 is also not a strong argument. There is ongoing research to solve such privacy concerns of memorizing training data. Also, decoding usually requires access to model weight, which is not often available for deployed ML systems as a server. Even keeping that argument aside,  by no means such an excuse allow for doing more privacy concerning developments.
    - I find Critique #3 even less intuitive. The example regarding K-nearest neighbours is also not completely valid, and it can't always memorize all data. On top of that, it's a classic algorithm. No recent methods on CIL use such methods. In modern machine learning, there is a **clear** distinction between a **model** and a **dataset**.
    - Under 2.2.2. Yes, it's a valid point that we want to reduce the compute cost as much as possible, but not at the expense of performance. After all, training happens once; deployment cost is what matters, and the argument has no indication of a solution to reducing that.
        - Edge devices usually delete the model, not train it - unless it's a federated learning system.

---

> ### Author Response · Authors · 2024-11-05
> **Response to Reviewer Ep3w**
>
> Thank you for taking the time to review the paper. We respond to specific feedback below.
>
> &nbsp;
>
> >If we remove the old restriction and we have all the previous data available, we do not need a specialized continual learning solution. We can simply rely on a supervised solution. The entire point of the literature and the existing methods is to make sure to learn from new data while not forgetting the old ones.
>
> As discussed in Section 2.2.2 (second paragraph), resource constraints make specialized continual learning solutions necessary. Memory and/or compute budgets disallow a "supervised solution" (i.e. trivial retraining on all data at each task). Also, we argue that "entire point of the literature and the existing methods" in CIL is *not* simply avoiding forgetting -- as discussed in Section 3, plasticity, across-task discrimination, and knowledge transfer are additional subproblems.
>
> &nbsp;
>
> >Again, Critique #1 is not a significant problem.
>
> The "Old Restriction: Unavailability of Previous Data" problem statement allows continual learning solutions which use more compute and memory than trivial retraining. We argue that this is a significant problem -- if a continual learning solution is less efficient than trivial retraining, then it is unlikely to be used in practice.
>
> &nbsp;
>
> >Also, the training compute time is less of a concern compared to inference compute time.
>
> While this may be a concern in some applications, this is not universally true. We discuss inference compute briefly in Section 4, though perhaps we should add further discussion. Thank you for bringing this up.
>
> &nbsp;
>
> >Critique #2 is also not a strong argument. There is ongoing research to solve such privacy concerns of memorizing training data. Also, decoding usually requires access to model weight, which is not often available for deployed ML systems as a server.
>
> We focus on what is stored on the server-side. If the end user only has access to model predictions and cannot access the training data or model weights, then many security/privacy concerns become irrelevant (i.e., the end used cannot access sensitive data). If storing previously learned sensitive data is disallowed _on the server-side_, but sensitive data can be recovered from trained models which are stored _on the server_, then there is a problem. This is what we are discussing in Critique #2.
>
> &nbsp;
>
> >I find Critique #3 even less intuitive. The example regarding K-nearest neighbours is also not completely valid, and it can't always memorize all data. On top of that, it's a classic algorithm. No recent methods on CIL use such methods. In modern machine learning, there is a **clear** distinction between a **model** and a **dataset**.
>
> [1] is a recent (2023) continual learning method which uses K-nearest neighbors. In addition, solutions to CIL should not be limited to modern machine learning -- if there is an old and/or simple method which works well, why not use it? Further, even if there is a clear distinction between a model and a dataset in the non-continual setting, this distinction may break down in the continual setting. For example, what is the model in a replay-based CIL method? Does the model include the replay buffer? [2] discusses how the model size plus the replay buffer size should be considered for fair evaluations of CIL methods.
>
> &nbsp;
>
> >Under 2.2.2. Yes, it's a valid point that we want to reduce the compute cost as much as possible, but not at the expense of performance.
>
> Again, while this may be true in certain applications, it is not universally true. Taking this argument to the logical extreme, why bother with continual learning at all? If we do not want to reduce compute at the expense of performance, then trivial retraining could be used.
>
> &nbsp;
>
> >For the remaining paper, there are no experimental results to compare different methods.
>
> [2] is a recent survey including comprehensive experimental results. Our paper is instead focused on the problem statement.
>
> &nbsp;
>
> Finally, while the resource-constrained setting has received relatively less attention, there is existing work justifying its usefulness -- see [1], [3].
>
> **We strongly feel that the arguments regarding the resource-constrained problem statement are a necessary component of the paper. We would be happy to add further explanation in Section 2.**
>
> &nbsp;
>
> [1] Ameya Prabhu, Zhipeng Cai, Puneet Dokania, Philip Torr, Vladlen Koltun, and Ozan Sener. "Online continual learning without the storage constraint." (2023).
>
> [2] Zhou, Da-Wei, Qi-Wei Wang, Zhi-Hong Qi, Han-Jia Ye, De-Chuan Zhan, and Ziwei Liu. "Class-incremental learning: A survey." IEEE Transactions on Pattern Analysis and Machine Intelligence (2024).
>
> [3] Kumar, Saurabh, Henrik Marklund, Ashish Rao, Yifan Zhu, Hong Jun Jeon, Yueyang Liu, and Benjamin Van Roy. "Continual learning as computationally constrained reinforcement learning." (2023).

---

> > ### Comment · Reviewer_Ep3w · 2024-11-05
> >
> > I would like to thank the authors for their quick response to the review. However, I have concerns about the response, along with the original concerns regarding the paper and its contribution.
> >
> > - If the memory and privacy constraint is completely removed, technically, there is no need for continual learning solutions. While expensive, any supervised solution can be used. By no means the supervised method have to be expensive or need to be trained from scratch. It could be fine-tuning a foundation model. Or fine-tuning the previously trained encoder. The CIL literature exists due to more constraints than the compute constraints.
> > - While few papers exist that focus on reducing resource constants, that does not justify changing the whole definition of CIL, as approached in this paper.
> > - Even if the redefinition of CIL is the main contribution, the paper must report the prior works in such matrics. Without that, it does not even answer the simplest of the questions, "Which is the best method to use if I have X, Y, or Z constraints in terms of compute, memory, GPU hours, and so on?"
> > - The stability-plasticity dilemma existed due to the unavailability of the previous samples. How does it change when we have all the samples available? I see zero issues with stability, just do a rehearsal on all the samples.
> > - In addition to that, all the discussion under section 3 is just the challenges of class-incremental learning. Everything here is already known. What is expected from this paper is a discussion on how such a problem is solved/modified/simplified/reduced with the change in the definition of the problem - when we don't have memory constant and can keep all the previous samples. No such insight is provided in this paper.
> > - Response regarding C#1 is unsatisfactory. As I mentioned, (1) there are efficient fine-tuning techniques that are less computer expensive. With all the data available, that is an option, even if not the best one. (2) if the inference speed is not changed, training cost is not the reason to choose a suboptimal model over the best-performing model, which is undoubtedly the one that uses all data and fine-tunes a foundation model with the SOTA method.
> > - Could you please elaborate on this regarding inference compute: "While this may be a concern in some applications, this is not universally true." - Any ML system is trained with the end goal of deployment, which continuously runs. What is an example where training cost is more important than its inference cost?
> > - "We focus on what is stored on the server-side." - I don't see any such "server-side" discussion in the main paper. Even with that, the fact does not change that it's a violation of the privacy concern for which the CL literature motivated not storing all the samples.
> > - As far as I am concerned, [1] is not a paper that has been accepted/published in any peer-reviewed venues. I don't find it logical to back a discussion based on such a paper.
> > - I agree that if an old model works better, we should certainly use it. But should we use it when it comes at the cost of privacy? The argument of this paper itself indicates that methods like [1] can memorize training data.
> > - [2] discusses how the model size plus the replay buffer size should be considered for **fair evaluations** of CIL methods. It does not say buffer is part of the model or the model itself. By no means does it indicate that there is no difference between what a model is and what a dataset is.
> > - Finally, I also feel that the arguments regarding the resource-constrained problem statement are necessary for CIL, but that does not mean we need to drop our privacy and other concerns regarding all the previous samples. Also, the definition and critique-based formulation of CIL in this paper is ill-justified. It does not provide any comparison in terms of performance or the computational requirement of an existing method.

---

> > > ### Author Response · Authors · 2024-11-20
> > > **Revisions and final response to Reviewer Ep3w**
> > >
> > > We have uploaded a revised version of the draft with changes listed in green text. The revised version includes discussion on related problem statements, a potential real-world example which fits within the resource-constrained setting, and describes approaches which address few-shot CIL and leverage pretraining. We respond to some of the reviewer's concerns below:
> > >
> > > &nbsp;
> > >
> > > >Even if the redefinition of CIL is the main contribution, the paper must report the prior works in such matrics. Without that, it does not even answer the simplest of the questions, "Which is the best method to use if I have X, Y, or Z constraints in terms of compute, memory, GPU hours, and so on?"
> > >
> > > As mentioned in our response to reviewer dnW6, we feel that fully answering this question may require multiple future papers and many empirical evaluations (i.e., evaluating existing methods under a wide range of memory and compute budgets, developing methods which jointly optimize for resource efficiency and performance, etc.). Our goal is to draw attention to the resource-constrained setting.
> > >
> > > &nbsp;
> > >
> > > >The stability-plasticity dilemma existed due to the unavailability of the previous samples. How does it change when we have all the samples available? I see zero issues with stability, just do a rehearsal on all the samples.
> > >
> > > To be clear: we are not arguing that all previous samples should always be stored. Memory constraints may disallow this.
> > >
> > > Even without memory constraints, compute constraints may place implicit restrictions on the amount of data used for rehearsal. Straightforward rehearsal on all of the samples may be computationally demanding, especially when many classes have been learned.
> > >
> > > &nbsp;
> > >
> > > >Could you please elaborate on this regarding inference compute: "While this may be a concern in some applications, this is not universally true." - Any ML system is trained with the end goal of deployment, which continuously runs. What is an example where training cost is more important than its inference cost?
> > >
> > > Continual learning applications may be one such example, as both training and inference continuously run. The details depend on the ratio of training data to inference data, as well as application-specific constraints.
> > >
> > > &nbsp;
> > >
> > > >I agree that if an old model works better, we should certainly use it. But should we use it when it comes at the cost of privacy? The argument of this paper itself indicates that methods like [1] can memorize training data.
> > >
> > > We are arguing that storing the previously learned samples may be no worse (with respect to privacy concerns) than storing a model from which previously learned samples could be recovered. In other words, if the training data can be at least partially reconstructed from a learned model, then storing the learned model may not respect privacy concerns. We agree that privacy concerns are important and acknowledge that there is work on privacy-preserving continual learning, however we question if the "unavailability of previous data" restriction alone truly respects these concerns.
> > >
> > > Again, we thank the reviewer for taking the time to review the paper and provide detailed feedback.

---

> ### Author Response · Authors · 2024-11-06
> **Response to reviewer Ep3W**
>
> In brief, we are not arguing for memory and privacy constraints to be removed. Nor are we arguing for solutions that violate privacy -cerntainly not in applications where data  privacy is a concern. Nor are we advocating for "changing the definition of CIL". It is really up to individual researchers to define what flavor of continual learning they want to address, why, and how. What we are advocating is a perspective on continual learning in resource constrained settings.  Such a setting can, in its most general form, accommodate privacy budget as a resource as well. While in "train once and use for ever" settings training cost may be less critical, that is not the case in continual learning setting where the model has to be retrained, fine-tuned, adapted, etc. on an ongoing basis.   With regard to "fine-tuning" techniques, we note that they come with no rigorous performance guarantees. We will save our detailed consideration of  the reviewer comments to our planned revision of the paper after all of the reviews are in. Thanks again for your feedback.

---

> ### Comment · Reviewer_Ep3w · 2024-11-06
>
> This response does not address most of my concerns and none of the main issues with this paper- it does not propose it for the first time, has technical issues, and doesn’t provide any insight, findings, recommendations, or comparisons.
>
> Nonetheless, the response avoided most of my comments, and the few responses they provided do not provide any justification, citation, or results to ground such claims.

---

### Review · Reviewer_dnW6 · 2024-11-07

**Summary Of Contributions:**

This review paper re-examines Class Incremental Learning (CIL), arguing for a problem definition centered around resource constraints (memory and compute) rather than more traditional focus on data availability. The setting with restricted access to past data seems unrealistic and can potentially lead to inefficient solutions.

From this perspective, they highlight several key challenges in CIL and defines a metric for each of them: Balancing stability and plasticity, Learning across-task discrimination and Transferring knowledge across tasks.

The paper proceeds to explore several naive approaches to tackle CIL and finally provides a literature review of papers while framed around naive approaches and challenges they above.

**Audience:**

Yes

**Broader Impact Concerns:**

No Broader Impact Statement is required as this is a review work.

**Claims And Evidence:**

Yes

**Requested Changes:**

- The disconnect between the resource-constrained framework (Section 3) and the review of existing methods (Sections 4-6) is a major weakness that needs to be addressed better in the paper.
- I would at least mention briefly other types of continual learning, such as task-incremental and domain-incremental learning. Also, in the era of LLM it is worth mentioning that there are papers on few-shot incremental learning where we need to learn a new class on top of large pre-trained based clasifier.
- I would suggest to the authors to provide more detailed examples of real world systems that fit the settings outlined in the paper and which of the reviewed method would be best to suit this problem.

**Strengths And Weaknesses:**

Strengths:
- The paper offers a novel practical perspective on CIL by focusing on resource constraints and highlighting the often-overlooked challenge of across-task discrimination. This reframing is valuable and can lead to more practical research directions.
- The paper is well-organized and clearly explains complex concept. The literature review seems complete and described thoroughly.
- The paper further defines challenges and metrics on how these challenges can be addressed and measured.

Weaknesses:
- There is a disconnect between between the proposed framework (resource constraints and novel metrics) and the subsequent discussion of existing CIL methods. Sections 4, 5, and 6 do not analyze existing methods within the context of the proposed memory/compute framework or utilize the novel metrics introduced in Section 3.
- The paper is positioned as a review and has no experiments, but it also proposes a novel way to look at a new problem and defines novel metrics without actually evaluating existing methods using them. I would suggest the author to answer the following questions in the introduction to make it really clear what is going on. What are the actionable lessons of the paper? Why do authors think that the audience would find this paper valuable? What are the claims of this paper?

---

> ### Author Response · Authors · 2024-11-13
> **Response to Reviewer dnW6**
>
> Thank you for the helpful feedback, and we apologize for the delay in response. We are currently revising the draft and will submit an updated version prior to the end of the discussion period.
>
> We will attempt to strengthen the connection between the resource-constrained problem statement and the later sections. This disconnect exists partly because most work on CIL does not address the resource-constrained setting. We hope that this paper will draw attention to the resource-constrained setting, and that future work will strengthen this connection. We feel that fully developing this connection may require multiple future papers and many empirical evaluations (i.e., evaluating existing methods under a wide range of memory and compute budgets, developing methods which jointly optimize for resource efficiency and performance, etc.).
>
> We will add a brief description other types of continual learning (TIL, DIL, etc.) in Section 2. The arguments supporting the resource-constrained problem setting extend beyond CIL -- we will add this discussion. Thanks for pointing this out. We will also add a subsection on pretrained models in Section 5.
>
> We can also provide examples of real-world applications which fit within the proposed setting. These include applications with high-volume, evolving data streams, which must be quickly and accurately classified -- for example, social media post classification, real-time fraud detection, and applications requiring user-specific personalization/specialization. We agree that this discussion would strengthen the paper.

---

> ### Author Response · Authors · 2024-11-20
> **Changes in revision in response to Reviewer dnW6**
>
> We have uploaded a revised draft with new content in green text. We highlight our changes in response to the reviewer's feedback below:
>
> - New subsection 2.4 describing a potential real-world example (real-time social media post classification) which fits within the resource-constrained setting, with discussion on potential solutions
> - New subsection 2.3 discussing related problem statements (few-shot CIL, TIL, DIL, online CL) and their corresponding challenges
> - Discussion regarding pretraining and few-shot CIL in section 5
> - New subsection 7.2 on actionable insights for CIL researchers
>
> We also include a table in section 1 summarizing existing reviews on CL/CIL as compared to our work, as well as further discussion in section 7 on future work.
>
> Again, thank you for helping improve this paper!

---

> > ### Comment · Reviewer_dnW6 · 2024-12-09
> >
> > The authors made all the requested changes to the paper, addressing all of the concerns that I had. I think that the paper is valuable and the community would benefit from having it published at TMLR.

---

### Decision · Action_Editor_vmjJ · 2025-01-22

**Recommendation:** Accept as is

**Comment:**

This survey reframes class-incremental learning (CIL) as learning to classify under resource constraints. Two reviewers found this perspective valuable, and the survey thorough, especially in its treatment of prior related work, including prior surveys. The remaining reviewer challenged the reformulation of CIL and viewed the submission as insufficient without any novel results. While indeed the survey doesn't present new results of its own, the majority of reviewers assessed that there is enough of a fresh perspective and targeted discussion of areas for future research for this survey to be a valuable contribution to TMLR, especially after the authors' revision to better connect the reformulation with existing work in CIL. I agree with this assessment and recommend acceptance.

**Audience:**

Yes.

**Claims And Evidence:**

Yes.